# Deep Learning Methods for Omics Data Imputation

**DOI:** 10.3390/biology12101313

**Published:** 2023-10-07

**Authors:** Lei Huang, Meng Song, Hui Shen, Huixiao Hong, Ping Gong, Hong-Wen Deng, Chaoyang Zhang

**Affiliations:** 1School of Computing Sciences and Computer Engineering, University of Southern Mississippi, Hattiesburg, MS 39406, USA; 2Center for Biomedical Informatics and Genomics, School of Medicine, Tulane University, New Orleans, LA 70112, USA; 3Division of Bioinformatics and Biostatistics, National Center for Toxicological Research, U.S. Food and Drug Administration, Jefferson, AR 72079, USA; 4Environmental Laboratory, U.S. Army Engineer Research and Development Center, Vicksburg, MS 39180, USA

**Keywords:** omics imputation, deep learning, multi-omics imputation

## Abstract

**Simple Summary:**

Missing values are common in omics data and can arise from various causes. Imputation approaches offer a different means of handling missing data instead of utilizing only subsets of the dataset. However, the imputation of missing omics data is a challenging task. Advanced imputation methods such as deep learning-based approaches can model complex patterns and relationships in large and high-dimensional omics datasets, making them an increasingly popular choice for imputation. This review provides an overview of deep learning-based methods for omics data imputation, focusing on model architectures and multi-omics data imputation. This review also examines the challenges and opportunities that are associated with deep learning methods in this field.

**Abstract:**

One common problem in omics data analysis is missing values, which can arise due to various reasons, such as poor tissue quality and insufficient sample volumes. Instead of discarding missing values and related data, imputation approaches offer an alternative means of handling missing data. However, the imputation of missing omics data is a non-trivial task. Difficulties mainly come from high dimensionality, non-linear or non-monotonic relationships within features, technical variations introduced by sampling methods, sample heterogeneity, and the non-random missingness mechanism. Several advanced imputation methods, including deep learning-based methods, have been proposed to address these challenges. Due to its capability of modeling complex patterns and relationships in large and high-dimensional datasets, many researchers have adopted deep learning models to impute missing omics data. This review provides a comprehensive overview of the currently available deep learning-based methods for omics imputation from the perspective of deep generative model architectures such as autoencoder, variational autoencoder, generative adversarial networks, and Transformer, with an emphasis on multi-omics data imputation. In addition, this review also discusses the opportunities that deep learning brings and the challenges that it might face in this field.

## 1. Introduction

Omics data imputation is a method employed to predict or fill in missing values in large-scale multi-omics data, such as genomics, transcriptomics, proteomics, and metabolomics data. Missing values are common due to measurement errors, poor sample quality, technology limitations, and data pre-processing steps. For example, high-throughput technologies may produce noisy, inaccurate measurements or degraded samples, and technological limitations may also result in missing data. Situations such as values that fail the quality control criteria during data pre-processing steps may cause missing values as well.

Take a gene expression dataset with some missing values as an example; it is necessary to fill in these missing values before downstream analysis. In this case, we can utilize an autoencoder, a specific type of deep neural network, to address this challenge. It consists of an encoder that takes the gene expression data with fake missing values as input and a decoder that is used to reconstruct the original data. Once this autoencoder model is trained, it is able to predict missing values based on learned patterns. In practical applications, implementing deep learning methods for omics data imputation involves meticulous preprocessing, thoughtful model selection, hyperparameter tuning, and rigorous validation, which will be discussed in more detail in Section 2.

The imputation of omics data has a variety of applications in the field of biology. According to the data types, it can be categorized as follows in Table 1.

To tackle the challenges posed by missing data in omics data analysis tasks, researchers have developed various types of imputation methods. These methods, along with their comparison, are summarized in Table 2.

The selection of an imputation method depends on the specific characteristics of the data and the goals of the analysis. In this review, we mainly focus on deep learning methods for omics data imputation, which represent new research advancements in this field.

## 2. Current Deep Learning Methods for Omics Data Imputation

Both supervised and unsupervised deep learning methods have proven to be effective in omics data imputation due to their capacity for end-to-end learning and the modeling of complex patterns. Among the deep learning models used for omics data imputation, the most widely adopted ones are the family of deep generative models. These models employ deep neural networks to learn intricate patterns and generate new data samples that are highly flexible and realistic. By utilizing multiple layers of representation, deep generative models aim to capture intricate features and dependencies within the data. Notable examples include autoencoder (AE), variational autoencoder (VAE), generative adversarial networks (GANs), and the Transformer model for generative sequential data, which have been successfully applied to various domains, such as image generation, machine translation, and omics data imputation. Additionally, convolutional neural networks (CNNs), recurrent neural networks (RNNs), and other deep learning models have demonstrated great potential in extracting features from omics data. However, these models have been presented in the literature in various ways, often specific to particular research problems. Therefore, there is a need to give a systematic review and make comparisons of these different deep learning methods for omics data imputation. In this review paper, we provide a comprehensive summary of the fundamental theory, present basic mathematical formulations, and use illustrative figures to depict the model structures. Our aim is to facilitate the use of deep learning models, particularly AE, VAE, GANs, and Transformer models, in the computational biology and bioinformatics community.

First, the selection of different model depends on the specific demands or criteria of the analysis, such as the type and size of the data, the complexity of the relationships between variables, and the model characteristics.

For example, AE models excel at learning intricate relationships within omics data and are relatively straightforward to train. However, they may be prone to overfitting, especially when employing complex models or working with limited datasets. Additionally, they can lead to less interpretable latent spaces. In contrast, VAEs model the latent space probabilistically, allowing for more meaningful interpretations and sample generation. VAEs use techniques such as Kullback–Leibler divergence to ensure that the latent space conforms to a known distribution, mitigating overfitting. VAEs have particular advantages when the interpretability or modeling of the probability distributions of the latent space is essential. VAEs can be particularly useful for transcriptomics data imputation due to their probabilistic nature in modeling the inherent uncertainty in gene expression levels. VAEs are valuable in integrating multiple omics data types, facilitating the learning of a shared latent space that captures the underlying relationships between different data modalities. Nevertheless, the complexity of training VAEs increases due to the inclusion of an additional loss term (KL divergence) and the need for latent space sampling during both the training and generation stages.

Compared to VAEs, GANs do not impose explicit assumptions about the data probability density, offering more flexibility in generating diverse samples. However, GANs are more suitable for generating high-quality and visually realistic images. They can be applied to image data, such as histology or microscopy images, or any omics data that can be organized in a 2D image format. However, training GANs is challenging, as they are susceptible to instability during training. Factors such as mode collapse, hyperparameter sensitivity, and ensuring a balance between the generator and discriminator networks all contribute to the intricacies of GAN training.

Transformer models incorporate an attention mechanism, which enables the understanding of extra-long sequences by allowing the model to assess the importance of each element in relation to every other element. This capability enables the model to capture complex relationships and dependencies even in sequences of considerable length, such as genomics and protein sequence data. They can process sequences in parallel, enhancing the training efficiency on modern hardware. Nevertheless, Transformer models can be computationally demanding, with the number of attention operations growing quadratically as the sequence length increases.

Other deep learning models, such as CNNs and RNNs, are also useful in omics data imputation. CNNs excel in capturing spatial patterns within image-based omics data and can serve as foundational components for other models. However, they may not be as effective in modeling sequential data and do not inherently support variable-length sequences. On the contrary, RNNs are well suited for tasks involving time series or sequences of varying lengths. Nonetheless, RNNs can encounter challenges related to vanishing or exploding gradients when dealing with excessively long sequences.

To provide a concise overview of the deep learning models applied in various omics data imputation tasks, Table 3 below summarizes the strengths and weaknesses of commonly used deep learning models, along with references to related studies across different omics data types.

This paper will review the aforementioned deep learning models, with a focus on the deep generative models, in the subsequent sections.

### 2.1. Autoencoder and Its Variants for Omics Imputation

Autoencoder (AE) [36] is a family of deep generative models designed to learn a compact representation of data and then use it to reconstruct the original data. This feature makes AE well suited for the imputation of missing values by reconstructing the data based on the observed values. An AE consists of two essential parts: an encoder, which is responsible for reducing the dimensionality of the input data to form an encoding or bottleneck layer, and a decoder, which reconstructs the original data from the encoding. The loss function of AE aims to minimize the discrepancy between the input and reconstructed data. A typical structure of an autoencoder applied in single-cell RNA-seq (scRNA-seq) gene expression data imputation is illustrated in Figure 1.

The training objective of an autoencoder is to minimize the reconstruction error, which quantifies the disparity between the original and reconstructed data. Consequently, an autoencoder acquires a representation of the data that reflects their fundamental characteristics. This learned representation can be applied for tasks such as dimensionality reduction, anomaly detection, and data imputation.

Autoencoders are widely adopted in omics data imputation. For example, AutoImpute [13] effectively imputes missing values while minimizing alterations to the biologically uninformative gene expression values. AutoImpute utilizes an overcomplete autoencoder to regenerate the imputed expression matrix X by taking the sparse gene expression matrix R as input. The loss function for AutoImpute is
$$min_{E,D}||R-D_{\sigma }{\left(E\left(R\right))|\right|}_{0}^{2}+\frac{\lambda }{2}\left({\left||E|\right|}_{F}^{2}+{\left||D|\right|}_{F}^{2}\right)$$
where E represents the encoder and D represents the decoder, ***R*** represents the gene expression matrix as input, λ represents the regularization coefficient, ${\left||\cdot |\right|}_{0}$ implies that the loss is calculated only for the non-zero counts present in ***R***, and σ represents the sigmoid activation function. Although AutoImpute recovers fewer missing values for genes compared to other methods, it performs better by ideally retaining most of the true zeros present in the data. As a result, it achieves better cell-clustering accuracy, variance stabilization, and cell-type separability [13].

ScScope [14] uses an autoencoder in a recurrent way to perform imputations on single-cell RNA-seq data iteratively. After handling the batch correction of the scRNA count data using the batch correction layer, the input goes through the encoder and decoder networks. Then, the output is processed through a self-corrected layer called the imputer, which generates the imputed data for specific cells and features. Finally, the imputed data and the batch-corrected input are fed into the autoencoder for another round of iteration to relearn an updated latent representation. ScScope’s architecture facilitates the iterative enhancement of the imputed output by employing a specified number of recurrent steps, denoted as T. It reduces to a conventional autoencoder when T is set to 1.

The deep count autoencoder network (DCA) [16] is an autoencoder-based model that uses three output channels to represent the parameters of a negative binomial noise model or a zero-inflation negative binomial noise model. The training objective is focused on minimizing the reconstruction error while considering the likelihood of the noise model’s distribution, rather than directly reconstructing the input data in an unsupervised manner to capture non-linear gene–gene dependencies.

AutoComplete [7] utilizes an autoencoder model to handle heterogeneous phenotypes consisting of continuous and binary-valued phenotypes. It reconstructs all phenotypes for each individual from a latent representation. The encoding stage masks missing entries and transforms observed phenotype values into a latent representation. This latent representation is transformed back into the input space to reconstruct all phenotypes. Imputed entries are obtained through a sigmoid function to support binary phenotypes. Additionally, AutoComplete implements a copy-masking simulation procedure to introduce realistic missingness patterns in the observed data by applying missingness patterns from other random individuals with a certain probability.

Denoising AutoEncoder (DAE) [37] is a variant of the autoencoder neural network designed to learn robust data representations in the presence of noise or missing values. DAE is a useful tool for denoising, imputation, and anomaly detection, where the input data are often corrupted with noise or missing values.

SAUCIE [15] is a highly parallelized and scalable deep neural network designed for comprehensive single-cell data analysis tasks. It utilizes the ability of DAE to learn deep representations of data, allowing it to perform various tasks on a unified data representation. SAUCIE employs tailored regularizations at different layers to reveal distinct data representations. These include batch correction, clustering, and denoising. The method uses a two-dimensional bottleneck layer for visualization, a novel MMD-based penalty for batch effect removal, and an 1D regularization for automated clustering. These regularizations balance against the reconstruction accuracy, ensuring meaningful solutions and denoised measurements. Such regularizations enhance the interpretability of the model’s output for downstream analysis.

### 2.2. Variational Autoencoder (VAE) and Its Variants for Omics Imputation

Autoencoder can be deterministic or probabilistic. Compared to vanilla autoencoder, variational autoencoder (VAE) [38] introduces a probabilistic approach to describing an observation in the latent space. In scenarios where VAE is employed, there is an assumption that the data are sampled independently from an underlying identical distribution. The motivating idea behind VAE is that we wish to model a specific distribution, namely the distribution of the latent variable *z* given some input *x*.

Using Bayes’ theorem, we can express this distribution as follows:$$p\left(z|x\right)=\frac{p(x|z)p\left(z\right)\;}{p\left(x\right)\;}=\frac{p(x|z)p\left(z\right)\;}{\int p(x|z)dz\;}=\frac{p\left(x,z\right)\;}{\int p(x|z)dz\;}$$

The problem is that the posterior p(z|x) is intractable because the marginal likelihood *p*(*x*) involves an integral over *z*. When *z* is a high-dimensional variable, the multivariate integration ∫p(x|z)dz becomes intractable [39].

The goal of the VAE method is to obtain a distribution q(z|x) that minimizes the Kullback–Leibler (*KL*) divergence KL(q(z|x)∥p(z|x)) to best approximate the distribution p(z|x). Using the definition of *KL* divergence, we have
$$\begin{array}{cc} KL(q\left(z|x\right)∥p\left(z|x\right)) & \equiv E_{z∼q}\left[\log\frac{q\left(z|x\right)}{p\left(z|x\right)}\right]\\ & =E_{z∼q}\left[\log q\left(z|x\right)\right]-E_{z∼q}\left[\log p\left(z|x\right)\right]\\ & =E_{z∼q}\left[\log q\left(z|x\right)\right]-E_{z∼q}\left[\log p\left(x,z\right)\right]+E_{z∼q}\left[\log p\left(x\right)\right]\\ & =\log p\left(x\right)-E_{z∼q}\left[\log\frac{p\left(x,z\right)}{q(z|x)}\right] \end{array}$$

If we define the second term of the right-hand side (RHS) as Evidence’s Lower Bound (ELBO), then we have
KL(q(z|x)∥p(z|x))=logp(x)−ELBO

Given that *KL* divergence is always non-negative, we have the following:logp(x) ≥ ELBO

As the name of ELBO suggests, ELBO can be a meaningful lower bound on the log-likelihood: both are negative, but ELBO is lower. Given x, the log evidence logp(x) is fixed, so minimizing KL(q(z|x)∥p(z|x)) is equivalent to maximizing ELBO. Thus, we can further derive ELBO as
$$\begin{array}{cc} ELBO & \equiv E_{z∼q}\left[\log\frac{p\left(x,z\right)}{q\left(z|x\right)}\right]\\ & =E_{z∼q}\left[\log\frac{p\left(x,z\right)p\left(z\right)}{q\left(z|x\right)p\left(z\right)}\right]{=E}_{z∼q}\left[\log\frac{p\left(x|z\right)}{q\left(z|x\right)p\left(z\right)}\right]\\ & =E_{z∼q}\left[\log p\left(x|z\right)\right]+E_{z∼q}\left[\log\frac{p\left(z\right)}{q\left(z|x\right)}\right]\\ & =E_{z∼q}\left[\log p\left(x|z\right)\right]-KL\left(q(z|x)∥p\left(z\right)\right) \end{array}$$

Apparently, ELBO is negative. VAE uses ELBO as its loss function. The first term of the RHS is called the reconstruction loss and the second term is the *KL* divergence. As the loss function for VAE, the reconstruction term measures how well VAE can reconstruct the input data. In contrast, the *KL* divergence term measures the divergence between the learned latent distribution and a prior distribution.

VAE uses two neural networks including an encoder and a decoder network to model the approximate posterior distribution q(z|x) and the conditional probability distribution p(x|z).

In VAE, the encoder network is responsible for parameterizing the approximate posterior distribution q(z|x), which represents the distribution of the latent space given the input data x. This is achieved by mapping the input data x to the parameters of the approximate posterior distribution. These parameters depend on the nature of the data being modeled, e.g., probabilities for a Bernoulli distribution or means and variances for a Gaussian distribution.

The decoder network, on the other hand, is responsible for generating the reconstructed output $\tilde{x}$, given a sampled latent code z. To generate the latent code z for the decoder, it involves sampling from a standard Gaussian distribution and then applying a transformation using the parameters of q(z|x), thus called the reparameterization trick.

The decoder network can be seen as modeling the conditional distribution p(x|z) implicitly. Although the decoder network does not explicitly output the parameters of the distribution, it learns to generate reconstructions of the input data x given a latent code z. During training, VAE jointly optimizes the encoder and decoder networks to optimize ELBO. ELBO is maximized by minimizing the reconstruction error between the input and reconstructed data while minimizing the *KL* divergence between the approximate posterior distribution and the prior distribution. The network structure of VAE for gene expression data imputation is illustrated in Figure 2. To backpropagate the gradients, the reparameterization technique [38] is needed. By training these two neural networks together, VAE learns to extract meaningful and compressed representations of the input data that can be used for generative modeling and other downstream tasks.

The second term of ELBO has a closed form as long as we assume that the prior p(z) belongs to a known distribution. In the classical VAE algorithm, the authors used Gaussian and binomial distributions as the priors. In the omics imputation problem, because dropout events are so common, both zero-inflated negative binomial (ZINB) and negative binomial (NB) distributions are widely used.

The closed form of the second term can be derived as below. Without simplicity, if the assumed prior distribution is multivariate Gaussian, then, for the latent feature i, we have the prior as
$$p\left(z\right)\sim N\left(\mu_{i},\sigma_{i}^{2}\right)$$

Because q(z|x) is the approximation of p(z), we naturally assume the same model for the approximate distribution:$$q\left(z|x\right)\sim N\left({\tilde{\mu }}_{i},{\tilde{\sigma }}_{i}^{2}\right)$$

Then, the second term,
$$\begin{array}{cc} -\mathrm{KL}\left(q\left(z|x\right)∥p\left(z\right)\right) & =E_{z∼q}\left[\log\frac{p\left(z\right)}{q\left(z|x\right)}\right]\\ & =E_{z∼q}\left[\log\frac{\frac{1}{\sqrt{2\pi \sigma_{i}^{2}}}exp\left(-\frac{{\left(x-\mu_{i}\right)}^{2}}{2\sigma_{i}^{2}}\right)}{\frac{1}{\sqrt{2\pi {\tilde{\sigma }}_{i}^{2}}}exp\left(-\frac{{\left(x-{\tilde{\mu }}_{i}\right)}^{2}}{2{\tilde{\sigma }}_{i}^{2}}\right)}\right]\\ & =E_{z∼q}\left[\log\frac{{\tilde{\sigma }}_{i}}{\sigma_{i}}-\frac{{\left(x-\mu_{i}\right)}^{2}}{2\sigma_{i}^{2}}+\frac{{\left(x-{\tilde{\mu }}_{i}\right)}^{2}}{2{\tilde{\sigma }}_{i}^{2}}\right]\\ & =\log\left(\frac{{\tilde{\sigma }}_{i}}{\sigma_{i}}\right)-\frac{1}{2\sigma_{i}^{2}}E_{z∼q}\left[{\left(x-\mu_{i}\right)}^{2}\right]+\frac{1}{2{\tilde{\sigma }}_{i}^{2}}E_{z∼q}\left[{\left(x-{\tilde{\mu }}_{i}\right)}^{2}\right]\\ & =\log\left(\frac{{\tilde{\sigma }}_{i}}{\sigma_{i}}\right)+\frac{1}{2}-\frac{1}{2\sigma_{i}^{2}}E_{z∼q}\left[{\left(x-\mu_{i}\right)}^{2}\right] \end{array}$$

The last term of the above equation’s RHS can be further simplified as
$$\begin{array}{cc} \frac{1}{2\sigma_{i}^{2}}E_{z∼q}\left[{\left(x-\mu_{i}\right)}^{2}\right] & =\frac{1}{2\sigma_{i}^{2}}E_{z∼q}\left[{\left(\left(x-{\tilde{\mu }}_{i}\right)+\left({\tilde{\mu }}_{i}-\mu_{i}\right)\right)}^{2}\right]\\ & =\frac{1}{2\sigma_{i}^{2}}E_{z∼q}\left[{\left(x-{\tilde{\mu }}_{i}\right)}^{2}+2\left(x-{\tilde{\mu }}_{i}\right)\left({\tilde{\mu }}_{i}-\mu_{i}\right)+{\left({\tilde{\mu }}_{i}-\mu_{i}\right)}^{2}\right]\\ & =\frac{1}{2\sigma_{i}^{2}}\left\{E_{q}\left[{\left(x-{\tilde{\mu }}_{i}\right)}^{2}\right]+2E_{q}\left[\left(x-{\tilde{\mu }}_{i}\right)\left({\tilde{\mu }}_{i}-\mu_{i}\right)\right]+E_{q}\left[{\left({\tilde{\mu }}_{i}-\mu_{i}\right)}^{2}\right]\right\}\\ & =\frac{1}{2\sigma_{i}^{2}}\left[{\tilde{\sigma }}_{i}^{2}+2\times 0\times \left({\tilde{\mu }}_{i}-\mu_{i}\right)+{\left({\tilde{\mu }}_{i}-\mu_{i}\right)}^{2}\right]\\ & =\frac{1}{2\sigma_{i}^{2}}\left[{\tilde{\sigma }}_{i}^{2}+{\left({\tilde{\mu }}_{i}-\mu_{i}\right)}^{2}\right] \end{array}$$

Then, we have
$$-\;\mathrm{KL}\left(q\left(z|x\right)∥p\left(z\right)\right)=\log\left(\frac{{\tilde{\sigma }}_{i}}{\sigma_{i}}\right)+\frac{1}{2}-\frac{1}{2\sigma_{i}^{2}}\left[{\tilde{\sigma }}_{i}^{2}+{\left({\tilde{\mu }}_{i}-\mu_{i}\right)}^{2}\right]$$

Moreover, when we take $\sigma_{i}=1$ and $\mu_{i}=0$, we have
$$-\;\mathrm{KL}\left(q\left(z|x\right)∥p\left(z\right)\right)=\frac{1}{2}\left[1+\log\left({\tilde{\sigma }}_{i}^{2}\right)-{\tilde{\sigma }}_{i}^{2}-{\tilde{\mu }}_{i}{}^{2}\right]$$

Since ${\tilde{\mu }}_{i}$ and ${\tilde{\sigma }}_{i}^{2}$ are the output of the encoder network, this is a closed form.

The computation of the reconstruction loss term is done by comparing the difference between x and the reconstructed output $\tilde{x}$ of the decoder network. The choice of the reconstructed loss function depends on the specific characteristics of the data and the objectives of the VAE model. Cross-entropy and MSE losses are commonly used as they provide straightforward measures of the dissimilarity between the input and the output of VAE. Cross-entropy loss is ideal for classification tasks with multiple classes or binary outcomes, particularly when models predict probabilities with softmax activation. Conversely, MSE loss is designed for regression tasks, predicting continuous values when linear activation functions are used at the output layer.

Single-cell variational inference (scVI) [17] aggregates biological variability, the library size, and the batch effect across similar cells and genes. It uses a VAE framework to approximate the distributions for the observed expression values. To accomplish this, scVI treats the raw count data $X_{n}$ as a sample obtained from a ZINB distribution conditioned on the library size scaling factor $\ell_{n}$, the gene expression level $Z_{n}$, and the batch ID $S_{n}$. The encoder uses two bottleneck networks as the approximation of the mean $\mu_{\ell }$ and variance $\sigma_{\ell }^{2}$ of the variational posterior $\ell_{n}=log\left(normal\left(\mu_{\ell },\;\sigma_{\ell }^{2}\right)\right)$, which is a cell-specific scaling factor that is strongly correlated to the log-library size. The encoder generates the latent space representation in parallel by reparametrizing the standard multivariate Gaussian prior $Z_{n}$ using two neural networks. The decoder uses two neural networks to approximate the ZINB distribution parameters. One is for approximating the NB distribution and the other one is used to approximate the zero-inflation probability [17]. Using this model, scVI learns a non-linear embedding of cells that can be used for multiple analysis tasks.

Single-cell ATAC-seq analysis via latent feature extraction (SCALE) [10] overcomes the increased sparsity of scATAC-seq data by using the Gaussian mixtures to model the latent prior and the VAE framework to approximate a more tightly estimated data distribution, thus achieving higher accuracy. By using the Jensen divergence distance between the distribution p of observed peaks over all samples and the distribution q of the predefined pattern for the cluster, SCALE defines the $score=1-\sqrt{Div_{jensen}\left(p,q\right)}$ as the prior of each Gaussian component that contributes to the latent space distribution.

### 2.3. Generative Adversarial Networks (GANs) and Their Variants for Omics Imputation

Generative adversarial networks (GANs) [40] include two key components, namely the Generator and the Discriminator. A Generator is trained to generate synthetic data samples that closely resemble real data, while a Discriminator is trained to differentiate between the real and the generated data. The training of GANs contains a series of alternating periods. First, the Discriminator network is trained for a certain number of epochs, during which it learns to distinguish between the real and the generated data. Then, the Generator network is trained for a specific number of epochs with the objective of generating synthetic data that can fool the Discriminator into classifying them as real. This process of alternating training between the Discriminator and Generator networks is repeated multiple times to refine the models and to improve their performance in generating realistic data, as illustrated in Figure 3.

In GANs, the Generator progressively improves its ability to generate synthetic data that closely resemble real data. Simultaneously, the Discriminator enhances its discriminatory power to accurately distinguish between the real data and the generated synthetic data. This dynamic interplay between the generator and discriminator networks leads to an adversarial training process, where both networks iteratively refine their respective capabilities. The loss function for GANs is similar to a min-max function in game theory [40]:$$\underset{G}{\min}\underset{D}{\max}\left[E_{x∼p_{\mathrm{data}}\left(x\right)}\left[\log\left(D\left(x\right)\right)\right]+E_{z∼p_{z\left(z\right)}}\left[\log\left(1-D\left(G\left(z\right)\right)\right)\right]\right]$$

In this loss function,

D(x) is the Discriminator’s estimation of the probability that real data instance x is real;

G(z) is the Generator’s output for given noise z;

D(G(z)) is the Discriminator’s estimation of the probability that a fake instance is real;

$E_{x∼p_{\mathrm{data}}\left(x\right)}$ is the expected value over all real data instances;

$E_{z∼p_{z\left(z\right)}}$ is the expected value over all generated fake instances G(z).

The formula is derived from the cross-entropy between the real and generated distributions. The Generator cannot directly affect the logD(x) term in the function, so, for the Generator, minimizing the loss is equivalent to minimizing log(1−D(G(z))).

In the phase of training the Discriminator, the real dataset and the Generator will provide training data for the Discriminator. The loss function of the Discriminator becomes
$$\underset{D}{\max}\left[E_{x∼p_{\mathrm{data}}\left(x\right)}\left[\log\left(D\left(x\right)\right)\right]+E_{z∼p_{z\left(z\right)}}\left[\log\left(1-D\left(G\left(z\right)\right)\right)\right]\right]$$

The objective of the Discriminator is to maximize the average log probability of real data x and minimize the log probability of fake data z. In this phase, the backpropagation process only goes through the Discriminator and updates the weights.

During the training phase for a Generator in a GAN, the backpropagation goes through both the Discriminator and the Generator, but only the weights of the Generator are updated. The Generator generates fake samples to fool the Discriminator using the following loss function:$$\underset{G}{\min}\left[E_{z∼p_{z\left(z\right)}}\left[\log\left(1-D\left(G\left(z\right)\right)\right)\right]\right]$$

During the training process, as the Generator improves its capacity to generate authentic data, the Discriminator’s performance deteriorates as it encounters difficulty in discerning between real and synthetic data. Once the Discriminator achieves accuracy of around 50%, it indicates that the Generator is well trained and can generate new data. As another family of deep generative models, GANs are also widely used in omics data imputation.

DeepHiC [1] is a conditional GAN-based method designed for the prediction of high-resolution Hi-C contact maps from low-resolution input data. By employing a CNN as the Generator, it generates high-resolution Hi-C contact maps, while the Discriminator network distinguishes between the generated maps and the real ones. To address the issue of over-smoothing caused by MSE loss, DeepHiC incorporates additional loss terms to preserve high-frequency information and a fine-scale structure in the predicted contact maps. These additional loss terms include perceptual loss, which captures structural features, and total variation loss, which suppresses artifacts. The enhanced Hi-C data generated by DeepHiC allow for the detection of chromatin loops similar to those found in deeply sequenced Hi-C data, surpassing the capabilities of the original low-coverage Hi-C data. It achieves a higher correlation and structural similarity index when comparing the enhanced data to the original high-resolution Hi-C matrices.

The training of GANs is challenging. As the Discriminator improves and becomes more accurate at distinguishing between real and generated data, the gradients backpropagated to the Generator may diminish, making it challenging to update the Generator effectively. This phenomenon is known as vanishing gradients and can hinder the learning process.

Wasserstein GANs (WGANs) [41] aim to address some challenges in traditional GANs’ training. WGANs utilize the Wasserstein distance as the loss function, which measures the similarity between the generated and real data distributions. Unlike traditional GANs, where the Discriminator outputs the probability of the input being real, WGANs employ a Critic network that provides a real-valued number representing the quality of the generated data. The Critic network is trained to approximate the true Wasserstein distance by minimizing a different loss function, rather than the loss function employed by the Discriminator in traditional GANs. As a result, WGANs offer improved stability and gradient flow during training, resulting in better convergence and data quality.

cscGAN [20] employs a conditional WGAN to impute missing values in single-cell RNA sequence data. The Generator consists of a fully connected network with three hidden layers that progressively increase in size. Each layer incorporates batch normalization, ReLU activation, and a library-size normalization (LSN) output layer. The inputs and outputs of the Generator represent single-cell expression levels resembling the training cells. The Critic is also composed of a fully connected network with three hidden layers that gradually decrease in size, employing ReLU activation.

scGAIN [18] adopts GAIN [42], a similar architecture to vanilla GANs, to solve the imputation problem of single-cell RNA sequencing data. However, the Generator network in GAIN is trained to impute only missing values in a dataset, rather than generating entirely new data samples. The input of the Generator is the concatenation of the original expression matrix with the missing value entries filled with random noise and the mask matrix. The Discriminator attempts to output a matrix similar to the “mask matrix”, distinguishing missing data (denoted by “0”) from observed data (denoted by “1”). Without providing any extra information, the Generator could output different imputation results each time with respect to the same Discriminator. Thus, to guarantee the quality of the imputed matrix, a hint mechanism is introduced by providing partial information about the masking to the Discriminator in the form of a “hint”. The hint mechanism chooses one element of each row of the mask matrix randomly and sets it to 0.5, which means that the corresponding entries could be either missing or observed data. Adjusting this value is equivalent to providing a different degree of known information from the mask matrix. This helps the Discriminator to focus on the given hints and improve the quality of the unknown ones. It is worth noting that scGAIN uses a different strategy from the original GAIN in training the model due to the highly zero-inflated characteristics of scRNA-seq data. During the initial epochs, genes with low expression are masked out, allowing the hint mechanism to choose genes with high expression to contribute to the loss function of the Discriminator. As the training goes on, more genes of low expression will be included to improve the accuracy.

Similar to scGAIN, GAIN-GTEx [33] uses the GAIN model to integrate numerical and categorical covariates’ embeddings from the GTEx dataset’s metadata for model training and outperforms other methods in both inductive and in-place imputation scenarios.

scIGANs [19] is a GAN-based model for single-cell RNA-seq imputation. It transforms real expression profiles into images for training. The Generator produces fake images representing gene expression profiles, while the Discriminator evaluates the authenticity. These networks compete to improve the performance. The generative model creates realistic scRNA-seq data for defined cell types, used to infer the true expression of dropouts. Using generated cells helps to avoid overfitting for common cell types and enhances the power for rare cells. The Generator can faithfully characterize cells for specific types, and a k-nearest neighbors (KNN) approach is used to impute dropouts in real scRNA-seq data of the same cell type. This method incorporates an equilibrium-enforcing technique [43] to ensure the balanced convergence of the Discriminator and the Generator during training.

### 2.4. Transformer and Its Variants for Omics Imputation

The Transformer model [44] is a cutting-edge language model renowned for effectively handling long-range dependencies between tokens, making it a state-of-the-art technique in natural language processing. This model has largely replaced the previous long short-term memory (LSTM) architecture [45] and demonstrated superior performance in various sequence-to-sequence mapping tasks. The Transformer is composed of two types of blocks, as depicted in Figure 4.

An encoder is responsible for building a representation of the input, and a decoder utilizes the encoder’s representation and other inputs to generate a target sequence, thus optimizing the model for output generation.

In this architecture, an encoder or decoder block is formed by stacking *N* identical building blocks. Each block consists of several sub-layers: a multi-head attention sub-layer and a position-wise, fully connected feed-forward layer [44]. The outputs of these sub-layers are concatenated with a residual connection and then passed through layer normalization. To facilitate the model’s ability to incorporate information from different representation subspaces simultaneously, the Transformer architecture introduces a multi-head attention mechanism. The multi-head attention layer is simply the mapping of the concatenation of the output of several self-attention calculations:$$multi-head\;attention=\left[Z^{\left(1\right)};Z^{\left(2\right)};\ldots ;Z^{\left(H\right)}\right]W_{o}$$

H is the number of the heads and the input X∈ℝn×d is interpreted as the length of n sequence of tokens, each with d features. In a self-attention calculation, the attention matrix Z(i) for each head i can be computed as
attention(Q(i),K(i),V(i))≡Z(i)=softmax(Q(i)K(i)Tdk)V(i)
where
Q(i)=XWQ(i), K(i)=XWK(i), V(i)=XWV(i)
WQ(i), WK(i)∈ℝd×dk, WV(i)∈ℝd×dv

$W_{Q}^{\left(i\right)},\;W_{K}^{\left(i\right)}$ and $W_{V}^{\left(i\right)}$ are initialized with random weights that can be updated in training, mapping $x_{i}$ from d to $d_{k}$ or $d_{v}$ dimensions ($d_{k}$ need not be equal to $d_{v}$, but usually it is convenient if we set $d_{k}=d_{v}=\frac{d}{H}$). Figure 5 provides a matrix-based visualization of the multi-head attention calculation process, offering insight into the dimensional changes as the input passes through this sublayer.

The add and norm layer within the Transformer model serves as an optimization technique with the objective of enhancing the training efficiency. By applying this layer, the model aims to streamline the learning process and expedite convergence during training, ultimately improving the overall efficiency of the training procedure.

Let x∈ℝd be the input token, μ∈ℝ the mean, and σ∈ℝ the standard deviation for each token; the layer normalization is computed as follows:norm=x−μσ+ϵ
where *ϵ* is a small constant used to avoid division by zero.

Diverging from the encoder’s self-attention layers, the self-attention layers in the decoder exhibit distinct characteristics. Specifically, the decoder’s self-attention layer exclusively attends to preceding positions within the output sequence. This is achieved through a masking mechanism that assigns a value of negative infinity to future positions prior to the softmax operation in the self-attention calculation:$$attention\left(Q,K,V\right)\equiv Z=softmax\left(\frac{QK^{T}}{\sqrt{d_{k}}}+M\right)V$$
so that the *i*-th attention vector $z_{i}$ for the *i-th* query is only related to the previous *i* steps. Accordingly, during the training process, the query of the decoder is required to only depend on previous information.

This look-ahead masking is required during training, but it is not required during inference. During inference, the decoder attends to the encoder output and generates the subsequent tokens in the output sequence based on the context provided by both the encoder output and the previously generated tokens.

Similar to the multi-head self-attention mechanism, the encoder–decoder cross-attention layer operates by constructing its Q (query) from the layer beneath it, while utilizing the K (key) and V (value) from the encoder stacks’ output. This enables the layer to effectively capture the interdependencies between the encoder and decoder representations during the information exchange process. The encoder–decoder architecture in Transformer models uses attention mechanisms that are not incorporated with positional information. Thus, to solve this, the Transformer simply uses

The embedding layer to encode the meaning of the word;The positional encoding layer to represent the position of the word;A combination of these two encodings by adding them to process the input ***X***.

Suppose that the uncoded input representation X∈ℝn×d contains the *d* dimension embeddings for n tokens of a sequence; for xi, the positional encoding vector pi is as follows:pi=PE(pos,dim)={sin(pos100002i/d)if dim=2icos(pos100002i/d)if dim=2i+1

Transformer-based models have been widely applied in omics imputation [8,12,34,46]. For example, DNA methylation entails the addition of a methyl group to DNA, predominantly occurring at CG dinucleotides, known as CpG methylation. However, the coverage of CpG sites per cell is constrained due to the limited genetic material per cell. Furthermore, the measurements of DNA methylation exhibit noise due to fewer reads at profiled sites. To overcome these challenges, CpG Transformer [12] adapts the BERT [47] architecture to directly process methylation matrices. By combining axial attention with sliding window attention, it enables the model to effectively capture interactions between neighboring CpG sites within and across cells, offering versatility in learning patterns. CpG Transformer consists of four Transformer blocks, each with three layers: row-wise sliding window eight-head attention, column-wise eight-head attention, and a position-wise fully connected feed-forward network. Add and norm layers are utilized between these layers. The sliding window row attention allows queries to attend to keys within a fixed window in the same row, while column attention enables queries to attend to keys across all elements in the same column. This methylome imputation model uses a 2D Cell-CpG site map (a 3D tensor) as input and is trained on the masked language modeling task [47], using cross-entropy loss to predict the original methylation state based on corrupted input.

Split-Transformer Impute (STI) [8] is an extended Transformer model tailored to genotype imputation, offering a reference-free approach applicable to diverse datasets with reduced preparation effort. STI utilizes cat-embedding for allele-specific representations and splits SNPs/SNVs into small windows to leverage the existing linkage disequilibrium (LD), resulting in computational savings and enhanced imputation quality. Each window is processed in a dedicated branch, including additional neighboring SNPs to preserve the accuracy at window borders. Attention blocks and convolutional blocks are employed to improve performance, with the final output assembled from concatenated branch outputs. STI’s haplotype-based imputation outperforms direct genotype encoding, leading to notable imputation accuracy improvements.

Other methods, although not directly conducting imputation tasks, can also be used to impute missing omics data. For example, Vaishnav et al. proposed a sequence-to-expression deep neural network model [34] based on the Transformer model to predict yeast gene (encoding the yellow fluorescent protein) expression from its promoter DNA sequence. This approach employs a CNN for feature extraction from one-hot-encoded DNA sequences. The extracted features are then passed through two Transformer encoder blocks with eight-head attention layers, followed by a bidirectional LSTM. The final step involves a fully connected layer for the prediction of gene expression. Žiga Avsec et al. proposed a portmanteau composed of an enhancer and transformer (Enformer) model [46] to predict gene expression and chromatin states in humans and mice.

### 2.5. Other Deep Learning Models and Their Applications in Omics Data Imputation

Convolutional Neural Network (CNN) [48]: CNNs are a family of neural networks that are designed to handle data with a grid-like structure, such as image data. In omics data imputation, the CNN has been used to impute missing values by learning patterns in the data based on the relationships between neighboring variables.

Sparse Convolutional Denoising Autoencoder (SCDA) [5] is designed specifically for genotype imputation. It integrates CNNs into an AE framework to capture the highly correlated LD and linkage patterns present in genotype data. Each convolutional kernel produces a feature map that incorporates LD patterns within the filtering window. Additionally, the SCDA model employs L1 regularization on each convolutional kernel to induce sparsity.

Xpresso [35] predicts mRNA abundance solely from promoter sequences based on a CNN. It applies Simulated Annealing (SA) and the Tree of Parzen Estimators (TPE) strategies to tune an optimal deep convolutional neural architecture.

DR Kelley [49] introduced a deep multi-task CNN that aims to learn the intricate regulatory grammars controlling transcription factor binding, chromatin marks, and transcription. Based on the underlying hypothesis that the inclusion of shared regulatory grammars among related species can significantly improve the accuracy of regulatory sequence activity models, this model utilizes genomic profiles from a diverse range of human and mouse cell types, enabling joint training on extensive data across multiple species. HiCPlus [2] uses CNNs to map a low-resolution interaction matrix to a high-resolution interaction matrix.

Recurrent Neural Network (RNN) [50]: RNNs are a family of neural networks that are designed to handle sequential data, such as time series data. In omics data imputation, RNNs have been used to impute missing values by learning patterns in the data based on the relationships between variables over time.

DeepCpG [11] employs a bidirectional RNN to analyze the neighboring methylation states of cells, alongside a CNN to process the DNA sequence. This combined approach enables the accurate inference of unmeasured methylation statuses at the individual cell level. To predict methylation states across all cells at target sites, the joint module leverages the learned interactions between the output features of the DNA and CpG modules through a multi-task architecture.

Additionally, numerous other studies have embraced the application of deep neural networks to impute missing values in omics data. For example, Avocado [9] transforms epigenomic data into a 3D tensor and performs factorization along three orthogonal latent factors: cell types, assay types, and the genomic axis. These three latent factors, representing the location of the training sample within the tensor, are concatenated and passed through a neural network consisting of two hidden dense layers to generate the ultimate prediction. DeepImpute [21] is a deep neural network model that imputes genes in a divide-and-conquer approach by constructing multiple sub-neural networks. TDimpute [51] is a neural network-based method that utilizes transfer learning to impute missing gene expression values in a multi-omics dataset. Initially, it trains a three-layer fully connected neural network using the pan-cancer dataset obtained from The Cancer Genome Atlas (TCGA) [52]. Subsequently, this network is fine-tuned on the specific cancer dataset, enhancing its performance and adaptability to a target dataset.

In summary, the integration of deep learning models to impute missing values in omics data represents a pivotal advancement in data analysis. These models demonstrate a capacity to effectively handle complex biological datasets, providing more accurate and reliable results for downstream analyses.

### 2.6. Evaluation Metrics for Model Comparison

When comparing different imputation models, selecting appropriate evaluation metrics is crucial to make informed decisions about model performance. The choice of different metrics depends on the specific task and goals of the models.

In cases where imputation is deemed as a classification prediction task, metrics such as accuracy, precision, recall, F_1_/F_2_ score, area under the receiver operating characteristic curve (AUC-ROC), and area under the precision–recall curve (AUC-PR) are the most common choices.

In most of the cases, such as imputing gene expression values, imputation tasks can be viewed as regression problems. In these scenarios, common evaluation metrics such as the coefficient of determination (R^2^) and normalized mutual information (NMI) can be used to evaluate the model’s performance.

In some domains, there may be specific metrics that are more meaningful. For example, ScScope [14] defines a metric called imputation accuracy, which is the normalized distance between the imputed log count entries and ground truth log count entries. STI [8] adopts both accuracy and a metric known as the imputation quality score (IQS) [53] to assess the effectiveness of models in genotype imputation. The model’s accuracy specifically pertains to missing positions, while IQS accounts for chance concordance between predicted and actual SNPs and is applicable to bi-allelic events. In Hi-C imputation, the structural similarity index (SSIM) [54] and the Jaccard index may be used as evaluation metrics to assess the quality and similarity of chromosomal contact maps.

Most research adopts multi-metric evaluations, which not only aid in mitigating the limitations of single-metric evaluations but also provide a more robust basis for the assessment of the overall performance and generalization capabilities of the regression model. For example, R^2^ is sensitive to the number of predictors and the possibility of overfitting. Prominent alternatives, such as the mean squared error (MSE), mean absolute error (MAE), and root mean squared error (RMSE), are widely utilized to gauge the predictive accuracy and residual errors. By integrating these complementary measurements alongside R^2^, a more comprehensive understanding of the model’s efficacy can be attained, enabling researchers and practitioners to make more informed decisions regarding model selection and refinement.

Furthermore, the evaluation metrics for downstream tasks such as clustering performance can also be used to justify the imputation quality. Clustering evaluation metrics such as the adjusted rand index (ARI) will be utilized when ground truth data are available. Conversely, in cases where the ground truth is unavailable, the Silhouette coefficient (SC) will serve as the designated metric.

These metrics offer distinct viewpoints on model performance, reflecting different aspects of imputation accuracy, stability, and robustness. It is important to note that many studies have utilized multiple metrics to provide a well-rounded evaluation. In order to comprehensively assess the effectiveness of deep learning-based methods across various imputation tasks and applications, we summarize the common performance evaluation metrics employed in these representative research endeavors in Table 4. This table does not serve as a direct performance comparison due to the varying applications and experimental conditions across different papers. The results listed in the table pertain only to specific data and experimental settings.

The comparative assessment of deep learning models necessitates a systematic and context-sensitive approach to metric selection. It is acknowledged that no universal evaluation criterion exists, prompting researchers to engage in thoughtful and discerning processes. Such processes entail meticulous consideration of the specific problem domains, the intended applications of the models, and the contextual relevance of evaluation metrics. By judiciously combining pertinent metrics and conducting a thorough analysis of their implications, researchers can effectively navigate the intricate landscape of model evaluation.

## 3. Deep Learning to Solve Multi-Omics Imputation Problems

Multi-omics analysis must handle highly heterogeneous data, including DNA sequences, raw counts, mass spectrometry intensity, and images. Multi-omics imputation refers to filling missing values in multi-omics data, typically including genomics, transcriptomics, epigenomics, proteomics, and metabolomics data. It is an essential step in analyzing multi-omics data and can lead to an improved understanding of the relationships between different omics data types and better insights into biological processes. Because of its ability to effectively deal with multi-modal data, deep learning is an ideal approach to incorporating multi-omics data into an integrated analysis framework.

scMM [55] is an autoencoder-based imputation method for the analysis of single-cell multi-omics data, allowing researchers to study individual cells’ characteristics across multiple modalities, such as gene expression, epigenetics, and surface proteins. It is a deep generative model-based framework that can extract low-dimensional joint representations from high-dimensional multi-modal data, making it easier to interpret the underlying biological relationships. One of its unique features is its ability to accurately predict missing modalities through cross-modal generation, meaning that scMM can generate data from one modality based on data from another. This makes it a valuable tool for the prediction of multiple missing modalities.

BABEL [56] is a deep learning model that incorporates single-cell data modalities to predict single-cell expression from scATAC-seq data and vice versa. It uses two interoperable encoders and decoders to project and infer RNA or ATAC outputs from a shared representation. The interoperability of the encoders and decoders allows for the computational synthesis of paired multi-omics measurements when only one modality is available. It learns to estimate true expression values by overcoming noise through a negative binomial distribution for RNA and a binarization for ATAC outputs.

totalVI [57] addresses the task of imputing missing protein data in a specific dataset by leveraging information acquired from observed proteins in other datasets. It employs VAE to perform an end-to-end analysis of paired transcriptome and protein measurements in single cells. By taking matrices of RNA and protein-unique molecular identifier counts as input, totalVI for imputation is facilitated. Additionally, optional categorical covariates such as experimental batch or donor information can be included to aid dataset integration.

GLUE [58] integrates unpaired single-cell multi-omics data using omics-specific autoencoders with graph-based coupling and adversarial alignment. GLUE addresses the challenge of bridging various omics-specific feature spaces by explicitly capturing their regulatory interactions. This approach ensures biologically intuitive and comprehensive integration, enabling a deeper understanding of complex biological systems. GLUE utilizes omics-specific VAEs to extract low-dimensional cell embeddings from individual omics layers. Despite potential variations in data dimensionality and the generative distribution across layers, the embedding dimension remains consistent. To establish links between the omics-specific data spaces, GLUE incorporates prior knowledge through a guidance graph, in which omics features act as vertices. GLUE employs a graph VAE to acquire feature embeddings from the guidance graph. These embeddings are utilized in decoders to reconstruct omics data by conducting inner product operations with cell embeddings. This strategy effectively establishes connections between the omics-specific data spaces, ensuring consistent embedding orientation. Finally, an omics discriminator aligns cell embeddings from different omics layers through adversarial learning.

Vaishnav et al. presented a novel deep neural network model [34], based on the Transformer architecture, that focuses on predicting the expression of genes in yeast. Specifically, their model utilizes the promoter DNA sequence to forecast the expression levels of the yellow fluorescent protein.

Other multi-omics imputation approaches include the following. Enformer is suitable for systematic application in fine-mapping existing genome-wide association studies and estimating regulatory activity across species, facilitating the study of cis-regulatory evolution [46]. ExpressionGAN utilizes a GAN to explore the intricate network of regulatory elements governing the expression of a specific gene. By considering the gene’s DNA sequence, its interactions with different factors, and targeting desired mRNA levels, ExpressionGAN generates regulatory DNA sequences that encompass the entire gene regulatory structure, including both coding and neighboring non-coding regions [59]. INTERACT [60] integrates CNN and Transformer models to predict the effects of genetic variations on DNAm levels in the human brain from local DNA sequences. cTP-net [23] adopts a transfer learning approach to impute surface protein abundances from scRNA-seq data using deep neural networks.

## 4. Opportunities and Challenges for Deep Learning-Based Multi-Omics Imputation

Deep learning methods, especially deep generative models, have shown promise in imputation tasks due to their ability to learn complex latent embeddings and capture relationships among heterogeneous data. However, using deep learning methods for imputation poses several challenges, such as handling missingness patterns and avoiding overfitting. Additionally, deep learning models might be too large to train. Despite these challenges, deep learning methods have the capacity to enhance the imputation accuracy and reduce bias in single/multi-omics data imputation.

### 4.1. Opportunities of Deep Learning Approaches

The use of deep learning methods to solve imputation problems in biology is an active area of research, and there are several opportunities that deep learning brings to this field. Some of them are the following.

Multi-modal imputation: many biological datasets consist of multiple modalities, such as genetic data, transcriptomics data, and epigenetic data. By taking advantage of the complementary information contained in each modality, deep learning might help to disentangle the complex relationships between these components and determine their effects on the primary outcome.End-to-end learning: in the genomics field, data pre-processing steps can be demanding in terms of time and susceptible to errors due to the diverse range of experimental data sources involved. This characteristic of end-to-end learning allows complex models to learn directly from raw data, eliminating the need for extensive manual feature engineering and pre-processing steps. By leveraging the capability of deep neural networks, complex tasks such as genomic sequence analysis, protein structure prediction, and disease classification can be integrated into a unified pipeline to enhance the predictive power of the models.Increased scalability: with the expanding size and complexity of biological datasets, the ability of deep learning models to scale effectively will become more crucial. In the future, there is potential for the emergence of models capable of efficiently handling large-scale biological data and more effectively conducting data imputation thanks to further advancements in computer hardware such as GPUs.

In addition to the advantages mentioned above, the other merits of deep learning methods include the better handling of high-dimensional data and discovering non-linear relationships.

In summary, using deep learning methods for imputation problems in biology is an active area of research with several potential benefits. Deep learning can help to disentangle complex relationships between multiple modalities in biological datasets, such as genetic data, transcriptomics data, and epigenetic data, and take advantage of the complementary information contained in each modality. It can also integrate imputation with clustering and visualization into a unified pipeline, which increases the predictive power and reduces the time-consuming and error-prone nature of data pre-processing. Additionally, deep learning models can handle large-scale biological data and impute data in a computationally efficient manner, while also better handling high-dimensional data and discovering non-linear relationships.

### 4.2. Challenges of Deep Learning Approaches

The challenging problems in omics data imputation existed before the era of deep learning, but deep learning approaches should also address them.

1.Distinguish between dropouts and the events of biological gene silencing.

scRNA-seq data are assumed to be zero-inflated. Such data consist of true zeros and dropout zeros (missing). The biologically driven zeros are referred to as the true zeros, while those that are technically driven (such as genes that express RNA, but the abundances are below the limit of detection of the instrument) are referred to as dropout zeros. Some methods, such as MAGIC [61] and DCA [16], treat all zeros as “missing data” and output a matrix in which every gene is expressed in every cell as if there are no biological zeros, while other methods, such as ALRA [31], SAVER [62], scImpute [30], and DrImpute [27], do distinguish biological zeros from missing values.

Three typical mechanisms cause missing data [63]:Missing completely at random (MCAR): the probability of having a missing value is independent of both the observed data and the missing data.Missing at random (MAR): the probability of having a missing value is independent of any missing values but may depend on observed covariates or values.Missing not at random (MNAR): the missing data are neither MCAR nor MAR. The missing data depend equally on the missing and observed values. In this scenario, handling the missing values is usually unattainable, as it depends on the unseen data.

In the context of scRNA-seq data, for example, the true zeros are considered to be MCAR, because the missing values are uniformly distributed across the dataset, while the dropout events are viewed as MNAR, because they are related to unmeasured technical factors such as batch effects and the sequencing depth. However, it is noteworthy that in other types of omics data, the nature and causes of missing values may differ, and the classification of missingness as MCAR, MAR, or MNAR may vary accordingly.

Taking missing data mechanisms into account when dealing with missing values is crucial to ensure unbiased results. For instance, maximum likelihood inference can be used when missing data are MCAR or MAR. However, imputation methods that consider missingness mechanisms are required when dealing with MNAR data, and deep learning-based imputation methods have shown promise in handling this.

Simple mean/median imputation tends to introduce bias in the mean estimation if the data are not MCAR [64]. KNN-based imputation, as a non-parametric method, performs better than simple imputation methods and helps to deal with all forms of missing data (MAR, MCAR, MNAR) [26]. Multiple imputation methods create several complete versions of the data by replacing the missing values with plausible data values. It is possible to perform multiple imputation when data are MNAR, but the limitation of multiple imputation is that the number of possible models for the missing data mechanism is infinite.

In statistical modeling, the concept of identifiability pertains to the ability to estimate a model’s parameters accurately and unambiguously based on the available data. An identifiable model ensures that the parameters can be estimated without uncertainty or multiple solutions, even when dealing with noise or measurement errors. While various deep generative methods have addressed the scenario of MNAR data, the identifiability of their models under MNAR conditions is often not guaranteed. An identifiable VAE model called GINA [65] provides sufficient conditions under which the ground truth parameters can be uniquely identified via maximum likelihood learning using observed data and might be the answer to applying deep generative models in omics data imputation.

2.Choose an appropriate data distribution.

Giving reasonable assumptions about the data distribution is critical to downstream analysis. Many omics datasets, such as single-cell RNA sequencing datasets, are of high sparsity and specifically manifested as zero inflation. Obviously, for the read count data of scRNA-seq, the default normal distribution prior assumption for most deep generative models is inaccurate. First, the normal distribution describes continuous data, while the reads count data are discrete; second, the values of the read count data are non-negative integers. Among all count models, the NB or ZINB regression model has been proven to be a better choice in describing scRNA-seq data.

The ZINB model is parameterized with three parameters: mean (μ), dispersion (*θ*), and dropout (*π*). In a VAE-based scRNA-seq imputation model applying the *NB*/*ZINB* distributions, the count for the *i*-th cell and *j*-th gene can be expressed as [16]:$$NB\left({\bar{X}}_{ij}^{count}|\mu_{ij},\theta_{ij}\right)=\frac{\Gamma \left({\bar{X}}_{ij}^{count}+\theta_{ij}\right)}{\Gamma \left(\theta_{ij}\right)\Gamma \left({\bar{X}}_{ij}^{count}+1\right)}{\left(\frac{\theta_{ij}}{\theta_{ij}+\mu_{ij}}\right)}^{\theta_{ij}}{\left(\frac{\mu_{ij}}{\theta_{ij}+\mu_{ij}}\right)}^{{\bar{X}}_{ij}^{count}}$$
$$ZINB\left({\bar{X}}_{ij}^{count}|\pi_{ij},\mu_{ij},\theta_{ij}\right)=\pi_{ij}\delta_{0}\left({\bar{X}}_{ij}^{count}\right)+\left(1-\pi_{ij}\right)NB\left({\bar{X}}_{ij}^{count}|\mu_{ij},\theta_{ij}\right)$$
where *i* and *j* represent cell and gene indices, and ${\bar{X}}_{ij}^{count}$ is computed using
$$\bar{X}=zscore\left(log\left(diag{\left(s_{i}\right)}^{-1}X+1\right)\right)$$
where X and “zscore” represent the raw count matrix and z-score normalization, and the size factor for every cell, $s_{i}$, is calculated as the total number of counts per cell divided by the median of total counts per cell. Since ${\bar{X}}_{ij}^{count}$ is z-scored, and $\Gamma \left({\bar{X}}_{ij}^{count}+1\right)$ is near 1 for any scaling count, this term is usually ignored. By denoting the output matrix of the encoder as E, the output matrix of the decoder as D, and the output matrix of the bottleneck layer as B, the model uses backpropagation to approximate the optimal matrix forms of the ZINB distribution’s dropout probability, mean, and dispersion $\hat{\Pi }$, $\hat{M},$ and $\hat{\Theta }$, respectively:$$\begin{array}{cc} E & =ReLU\left(\bar{X}W_{E}\right),\;B=ReLU\left(EW_{B}\right)\\ D & =ReLU\left(BW_{D}\right),\;M=diag\left(s_{i}\right)\times exp\left(DW_{\mu }\right)\\ \Theta & =exp\left(DW_{\theta }\right),\;\Pi =sigmoid\left(DW_{\pi }\right)\\ \hat{\Pi },\;\hat{M},\;\hat{\Theta } & =argmin_{\Pi ,M,\Theta }\overset{n}{\underset{i=1}{\sum }}\overset{p}{\underset{j=1}{\sum }}-log\left(ZINB\left({\bar{X}}_{ij}^{count}|\pi_{ij},\mu_{ij},\theta_{ij}\right)\right)+\lambda \pi_{ij}{}^{2} \end{array}$$

λ is a tunable zero-inflation regularization parameter that acts as a prior on the weight of the dropout process.

Some research [17] suggests that methods based on a ZINB distribution perform better than those using NB distributions because it captures important aspects of technical variability for scRNA-seq data. While certain studies [66] assume the presence of zero inflation caused by technical artifacts and advocate for the suitability of a ZINB model, other recent studies [66] argue that although zero inflation is evident in scRNA-seq data, the NB distribution is a more appropriate choice. The debate about which model is more suitable is still inconclusive.

3.Release the power of multi-omics data imputation.

Different omics datasets, including genomics, epigenomics, transcriptomics, proteomics, and phenomics, have distinct biological roles and mechanisms. Despite the availability and diversity of numerous omics datasets, challenges persist in their acquisition, processing, efficient integration, and interpretation. As a result, it is arguably not possible to use a single deep learning framework to extract specific features from all these datasets [67]. In the realm of multi-modal data integration, significant strides have been made with the emergence of large language models. This advancement shows the potential to integrate various omics data into a unified framework [68].

4.Reproducibility and cost.

The computational cost of training numerous deep learning models using stochastic gradient descent can be substantial. The initial parameters for the model and the random seeds impart essential influences on the final performance, further complicating the reproducibility. Furthermore, the expense of creating a cutting-edge deep learning model from scratch can be enormous. Models with large capacities, such as Transformer-based models, require enormous data and computational power and can have more than one billion trainable parameters. According to estimates, using publicly available cloud computing resources, reproducing one of those models could cost millions of dollars.

5.Improve the interpretability of deep learning models.

While deep learning models can achieve high accuracy, they can sometimes be difficult to interpret. Future trends may see the development of more interpretable models, allowing biologists to better understand the underlying mechanisms that drive the imputation process.

6.Conclusions

As the most successful technique in the last decade, deep learning has been applied in predicting or imputing missing values in omics data and has achieved state-of-the-art performance. This review provides an overview of omics data imputation using deep learning approaches, specifically the mainstream of deep generative models and their applications in imputing various omics data. The review also covers multi-omics imputation using other deep learning models.

Deep learning methods for omics data imputation have achieved success due to several factors. One key factor is its inherent end-to-end learning capability, which eliminates the need for time-consuming and error-prone data pre-processing steps, particularly in genomics, where diverse experimental data sources exist. By integrating multiple pre-processing steps into a single model, deep learning eliminates biases and batch effects, enhancing the predictive power of models. Another factor is deep learning’s effectiveness in handling multi-modal data, as it can seamlessly integrate diverse types of data for joint analysis, including multi-omics data. Crucially, deep learning excels at handling large-scale and high-dimensional datasets. Recent advancements such as model parallelism and the potential for larger models have attracted researchers in bioinformatics to make a transition from traditional methods to deep learning-based omics imputation.

## Figures and Tables

**Figure 1 biology-12-01313-f001:**
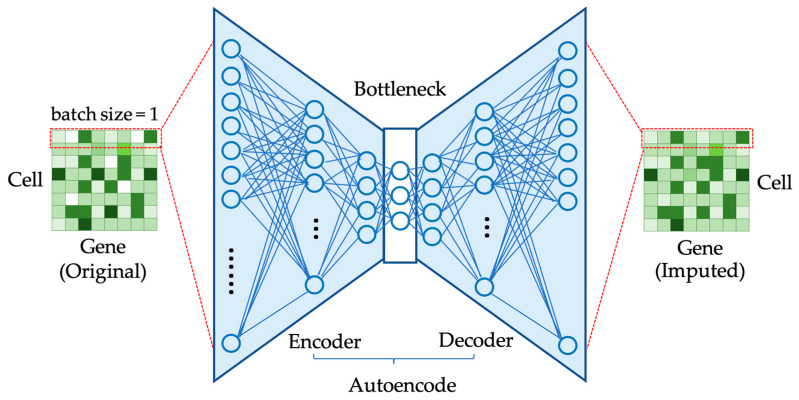
Single-cell RNA-seq data imputation using an autoencoder. Utilizing an autoencoder model, this figure demonstrates the imputation process for missing values in a 2D single-cell gene expression dataset. The various shades of green in the gene expression matrices indicate distinct levels of gene expression, while areas with missing data are displayed in white. A batch of the input samples is mapped individually to a compressed latent space and then reconstructed to generate a complete output, guided by a specialized loss function to ensure accuracy.

**Figure 2 biology-12-01313-f002:**
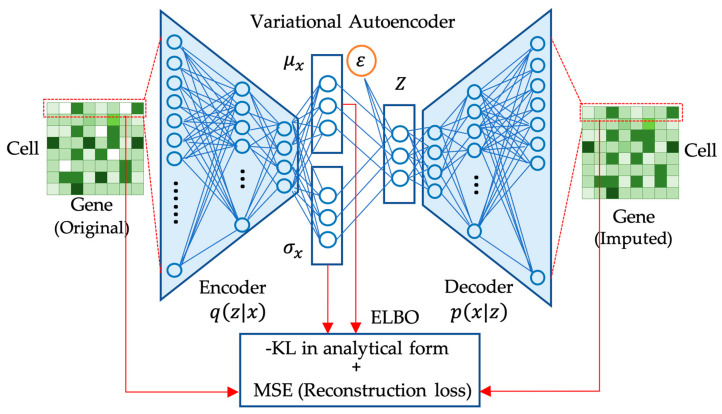
Variational autoencoder (VAE) imputation of scRNA-seq data. This diagram shows the VAE network structure, which employs mean squared error (MSE) reconstruction and KL divergence regularization to model latent feature distributions, ensuring accurate reconstruction while aligning with a Gaussian prior for meaningful interpretations and generative capabilities.

**Figure 3 biology-12-01313-f003:**
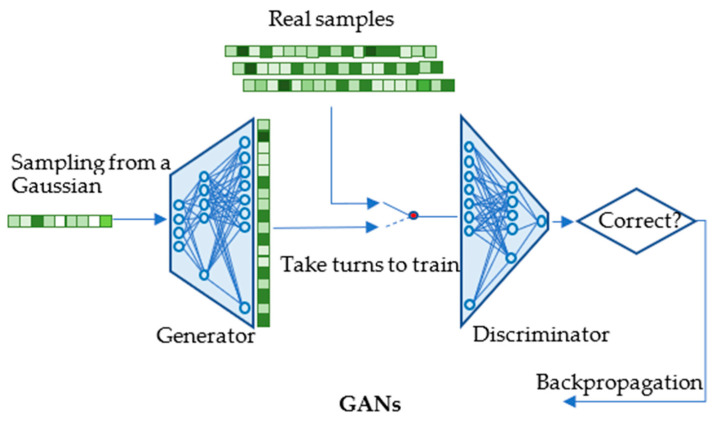
A typical structure and training process of generative adversarial networks (GANs). This figure illustrates how GANs are trained on single cell RNA gene expression samples. GANs consist of two key components: the Generator, responsible for creating synthetic data, and the Discriminator, which distinguishes between real and generated data. The training process involves alternating processes between these components, with the Discriminator learning first, followed by the Generator, in order to iteratively refine the models. In this way GANs learn to generate realistic samples.

**Figure 4 biology-12-01313-f004:**
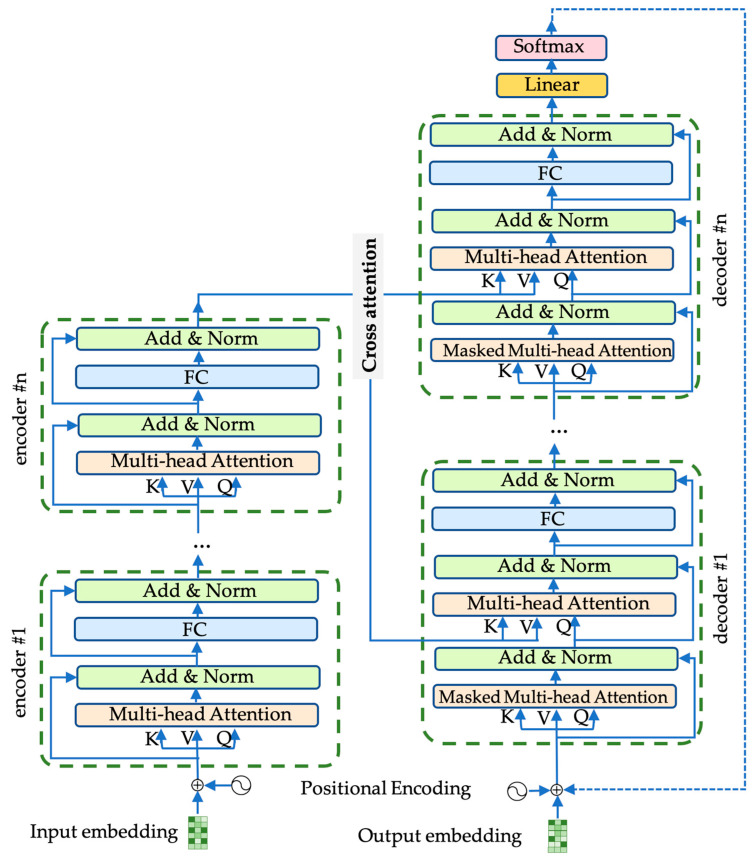
The Transformer architecture. This diagram illustrates the architecture of the Transformer model, featuring duplicated instances of two core blocks: encoder blocks and decoder blocks. These blocks incorporate essential components, such as multi-head attention/masked multi-head attention layers, fully connected layers, add and norm layers, and residue connections. When the batch size is set to 1, both the input and output embeddings can be visualized as matrices. In this representation, the rows correspond to the sequence length, the columns represent the embedding dimension, and varying shades of green indicate distinct values.

**Figure 5 biology-12-01313-f005:**
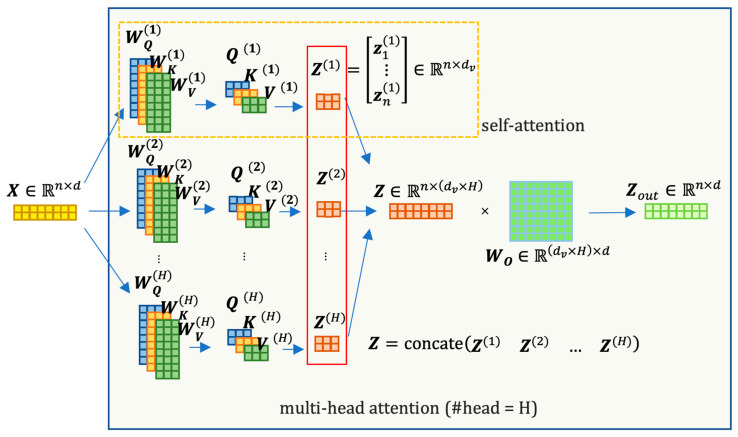
Multi-head attention layer calculation with illustrated dimensions. As depicted, the multi-head attention sublayer preserves the shape of the input data.

**Table 1 biology-12-01313-t001:** Omics imputation for various data types.

Data Type	Data Description	Causes of Missing Values	Imputation Method
Genomics data	Sequencing data: raw DNA sequence Hi-C data, typically represented as a Hi-C contact map that quantifies the intensity of physical interaction between two genome regions at the DNA level. The entries in the matrix are contact counts between pairs of genomic loci.	The sequencing depth and the complexity—for example, repetitive sequences or structural variations, which result in gaps or ambiguities in the sequencing data and dropouts in the Hi-C contact map.	Deep learning-based methods [1,2,3,4]
Genotype data	Genetic variation matrix, with each column representing an individual sample and each row representing a specific genetic marker or variant. The values indicate the alleles or genotypes present at those markers in the respective samples.	Low percentage of genotype calls or underrepresentation of rare variants.	Deep learning-based methods [5,6,7,8]
Epigenetic data	Include DNA methylation patterns, chromatin accessibility data (ATAC-seq), and histone modification profiles.	Technical limitations, cellular heterogeneity, and biological variability	Computational methods [9,10,11,12]
(Single-cell) RNA-seq data	Gene expression profile matrix, where each row corresponds to a sample (cell) and each column corresponds to a different gene, with the values indicating the expression levels of those genes in the respective samples (cell).	Low reverse transcription efficiency in single-cell RNA-seq data.	[13,14,15,16,17,18,19,20,21]
Proteomics data	Comprise essential information extracted from mass spectrometry results, including partial peptide sequences, peptide and protein identifications, modification profiles, quantification, and pathway-level insights. These data offer a detailed view of protein compositions, modifications, and abundance within a sample.	Imperfect identification of coding sequences and technology sensitivity limitations.	Statistical methods [22,23], deep learning-based methods [24]
Metabolomics data	Annotated feature matrix where each feature represents a specific signal or peak corresponding to a chemical compound. It indicates the relative abundances of these signals for each sample.	Experimental limitations, technical issues, and biological variability.	Statistical methods [25,26]

**Table 2 biology-12-01313-t002:** Comparison of imputation methods.

Method	Description	Pros and Cons	Application
Mean/Median Imputation	Substitutes missing values with mean or median values.	Pros: easy to implement. Cons: ignores relationships between variables, can introduce bias.	Used as baseline method
Hot-Deck Imputation	Employs clustering to find similar cells and then imputes missing values by copying values from identified donors. Multiple imputation rounds with varied cluster sizes ensure robustness, and final imputed values are averaged across all imputations.	Pros: uses similarity for imputation, potentially more accurate than mean imputation. Cons: requires identification of similar cases, may not always be feasible or accurate.	[27]
Multiple Imputation	Uses statistical methods to impute missing values and generates multiple imputed datasets through iterative rounds, and then combines results using Rubin’s Rules.	Pros: accounts for uncertainty in imputations, provides valid statistical inferences. Cons: more computationally intensive, requires specification of imputation models.	[28,29]
Classical ML Methods	Utilize classical machine learning techniques for omics data imputation.	Pros: can capture complex relationships between variables, potentially accurate imputations. Cons: requires careful model selection and validation, may overfit noisy data.	Random forest [25]; regression [30]; low-rank matrix representation /SVD [31]; KNN [32]
Deep Learning Methods	Leverage deep learning models to capture complex patterns and relationships in large and high-dimensional datasets.	Pros: can capture intricate patterns in high-dimensional data, potentially accurate imputations. Cons: require large amounts of data, computationally intensive, may be challenging to interpret.	Autoencoder [14]; variational autoencoder [17]; generative adversarial networks [1]; Transformer [8]

**Table 3 biology-12-01313-t003:** Commonly used deep learning models for omics data imputation.

Deep Learning Model	Strengths	Weaknesses	Application
AE	Ability to handle high-dimensional data with non-linear patterns	Potential for overfitting/limited interpretability	scRNA-seq [13,14,15,16]; phenotype [7]
VAE	Explicit modeling of the probability distribution	Challenging to train compared to AE	scRNA-seq [17]; scATAC-seq [10]
GANs	No explicit data probability assumption, can handle 2D data	Potential instability during training	bulk RNA-seq [33]; Hi-C [1]; scRNA-seq [18,20]
Transformer	Ability to model sequential data	Could be computationally expensive	bulk RNA-seq [34]; DNA methylation [12]; Genotype [8]
CNN	Excellent at capturing spatial patterns in image-based omics data, can be the building blocks of other models	Unable to effectively model sequential data and support variable-length sequences	bulk RNA-seq [35]; Genotype [5]; Hi-C [2]
RNN	Suited for tasks that involve time series or sequences of varying lengths	Vanishing or exploding gradients in dealing with excessively long sequences	DNA methylation [11]

**Table 4 biology-12-01313-t004:** Performance evaluation metrics used by representative deep learning-based methods with respect to different types of omics data. “-” means that the metric is not employed; “*” means that the metric is employed without reporting the exact value (only in visualization). All numerical values have been rounded to three significant figures.

Method	Model	Year	Data Type	Dataset	Performance Evaluation Metric						
					*r*	*R* ^2^	AUROC	ACC	MSE/ RMSE	ARI/ NMI	Other
**AE**	AutoImpute [13]	2018	scRNA-seq	Jurkat-293T	-	-	-	-	1.19/ RMSE	-	-
	ScScope [14]	2019	scRNA-seq	Retina	-	*	-	-	-	*/ ARI	0.3616/ Median L_1_ distance for all dropouts
	AutoComplete [7]	2022	Phenotype	Cardiometabolic/UKBB	-	0.263	0.739	-	-	-	-
**VAE**	scVI [17]	2018	scRNA-seq	CORTEX	-	-	-	-	-	*	2.45/ Median L_1_ distance for all dropouts
	SCALE [10]	2019	scATAC-seq	Leukemia	-	-	-	-	-	0.478 /ARI	-
**GANs**	GAIN-GTEx [33]	2021	bulk RNA-seq	GTEx	-	0.638	-	-	-	-	-
	DeepHiC [1]	2020	Hi-C	K562/ENCODE	*	-	0.825	-	-	-	*/Jaccard Index, SSIM
	cscGAN [20]	2020	scRNA-seq	PBMC	-	-	-	-	-	-	0.61/ ROC for a trained classifier
**Transformer**	CpG Transformer [12]	2022	DNA methylation	GSE65364	-	-	0.980	-	-	-	-
	STI [8]	2023	Genotype	1kGP	-	-	-	0.964	-	-	0.878/IQS
**CNN**	Xpresso [35]	2020	bulk RNA-seq	K562/ENCODE	-	0.61	0.65	-	-	-	-
**RNN**	DeepCpG [11]	2017	DNA methylation	GSE65364/GEO	*	-	*	*	-	-	-

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
