# Peer review of "Deep Learning Methods for Omics Data Imputation"

_biology, 2023, doi:10.3390/biology12101313_

Round 1

Reviewer 1 Report

1. Comparison table of literature works related to usage of deep learning methods for omics data imputation needs to included. 

2. That table may include the performance metrics comparison also. 

3. Advantages and limitations of deep learning methods for omics data imputation can be tabulated 

Reviewer 2 Report

Imputing missing value in omics data is essential for further analysis. In this manuscript, the authors reviewed deep learning methods used in omics data imputation field. This manuscript provides a comprehensive overview of the currently available deep learning-based methods for omics imputation and discusses the opportunities deep learning brings and the challenges deep learning methods might face in this field. I have several comments to improve this manuscript.

Major concerns:

Considering some potential readers(e.g. biologist) are not very expertise in this field, if possible, it is better to provide some toy examples to implement omic data imputation by using deep learning methods. Thus, they can better understand this review.

Minor comments:

Line47-115, if possible, please use a table to present these contents

In 2. Deep learning models currently adopted in omics data imputation section, it is better to use a table to summary these methods

In “3. Deep learning for solving the multi-omics imputation problem” section. This section looks like just list some methods. Please add some compare comments among these methods and point their advantages and disadvantages in imputating different omic data.

Reviewer 3 Report

The authors provide a comprehensive overview of the currently available deep learning-based methods for omics imputation from the perspective of deep generative model architectures such as autoencoder, variational autoencoder, generative adversarial networks and transformer, with an emphasis on multi-omics data imputation.  Albeit, I consider these findings to provide new insight into deep learning-related fields, I still have some suggestions.

1, Most figures are highly professional; however, the authors should guide the readers to the meaning of the images appropriately; otherwise, it will likely cause misunderstandings. Therefore, I suggest the author consider revising these figure legends again.

2, This review discusses the opportunities deep learning brings and the challenges deep learning methods might face in this field. However, it would be much better if the authors could provide some Workflow or Scheme for this research, I suggest that they can take a look at the recent paper in MDPI (PMID:  35563422, 36677020, 34834441)

3, There are few typo issues for the authors to pay attention to; please also unify the writing of scientific terms. “Italic, capital”? Please double-check superscripts and subscripts for the whole manuscript.

4, Most references are out of date, the author needs to discuss the recent paper as well as the analysis methods in this review.  

5, The font is too small for some of the current figures; meanwhile, the manuscript also needs English proofreading.

 Minor editing of English language required

Round 2

Reviewer 1 Report

I appreciate authors for incorporating the changes.

Except second comment, all other suggestions are incorporated. Even tough there is no benchmark datasets available for this work, at least performance metrics of some of the significant works can be included to provide more information to readers. Try to mention the data they used and performance metrics they attained. 

Author Response

Thank you for your constructive suggestion, we agree with your comments and we have add some words and Table 4 at the 2.6 subsection: "Evaluation metrics for models comparison", we also include two metrics (SSIM) and Jaccard index, and add respective reference. We highlighted the modification in the manuscript.
